# IRE1α regulates macrophage polarization, PD-L1 expression, and tumor survival

**Alyssa Batista**[1☯], **Jeffrey J. Rodvold**[1☯], **Su Xian**[2], **Stephen C. Searles**[1], **Alyssa Lew**[1],
**Takao Iwawaki**[3], **Gonzalo Almanza**[1], **T. Cameron Waller**[2], **Jonathan Lin**[4],
**Kristen Jepsen**[5], **Hannah Carter**[2], **Maurizio Zanetti**[1]*

**1** The Laboratory of Immunology, Department of Medicine and Moores Cancer Center, University of
California, San Diego, La Jolla, California, United States of America, **2** Division of Medical Genetics;
Department of Medicine, and Bioinformatics and Systems Biology Program, University of California San
Diego, La Jolla, California, United States of America, **3** Laboratory for Cell Recovery Mechanisms, Brain
Science Institute, RIKEN, Hirosawa, Japan, **4** Department of Pathology, Stanford University, Palo Alto,
California, United States of America, **5** IGM Genomics Center, University of California, San Diego, La Jolla,
California, United States of America

☯ These authors contributed equally to this work.
* mzanetti@ucsd.edu

## Abstract

In the tumor microenvironment, local immune dysregulation is driven in part by macro-
phages and dendritic cells that are polarized to a mixed proinflammatory/immune-suppres-
sive phenotype. The unfolded protein response (UPR) is emerging as the possible origin of
these events. Here we report that the inositol-requiring enzyme 1 (IRE1α) branch of the
UPR is directly involved in the polarization of macrophages in vitro and in vivo, including the
up-regulation of interleukin 6 (IL-6), IL-23, Arginase1, as well as surface expression of CD86
and programmed death ligand 1 (PD-L1). Macrophages in which the IRE1α/X-box binding
protein 1 (Xbp1) axis is blocked pharmacologically or deleted genetically have significantly
reduced polarization and CD86 and PD-L1 expression, which was induced independent of
IFNγ signaling, suggesting a novel mechanism in PD-L1 regulation in macrophages. Mice
with IRE1α- but not Xbp1-deficient macrophages showed greater survival than controls
when implanted with B16.F10 melanoma cells. Remarkably, we found a significant associa-
tion between the IRE1α gene signature and *CD274* gene expression in tumor-infiltrating
macrophages in humans. RNA sequencing (RNASeq) analysis showed that bone marrow–
derived macrophages with IRE1α deletion lose the integrity of the gene connectivity charac-
teristic of regulated IRE1α-dependent decay (RIDD) and the ability to activate *CD274* gene
expression. Thus, the IRE1α/Xbp1 axis drives the polarization of macrophages in the tumor
microenvironment initiating a complex immune dysregulation leading to failure of local
immune surveillance.

## Introduction

Myeloid cells in the tumor microenvironment (TME) are of central relevance to understand
the dynamics of tumor progression [1]. They infiltrate tumors in varying numbers depending

doi.org/10.1371/journal.pbio.3000687

of Edinburgh, UNITED KINGDOM

**Data Availability Statement:** All RNASeq data have
been deposited in BioProject database (accession
no. ID PRJNA622650; http://www.ncbi.nlm.nih.
gov/bioproject/622650).

**Funding:** NIH grant # RO1 CA220009 was given to MZ and HC. Frank H. and Eva B. Buck Foundation funding was given to JJR. Research reported in this publication was supported in part by the National Cancer Institute of the National Institutes of Health under Award Number T32CA121938 to SCS. The funders had no role in study design, data collection and analysis, decision to publish, or preparation of the manuscript.

**Competing interests:** The authors have declared that no competing interests exist.

**Abbreviations:** 4HNE, 4-hydroxynonenal; AIC, Akaike information criterion; Apc, adenomatous polyposis coli; Arg1, Arginase1; ATF, activating transcription factor; BMDC, bone marrow–derived dendritic cell; BMDM, bone marrow–derived macrophage; CHOP, CCAAT-enhancer-binding protein homologous protein; CKO, conditional knock-out; CM, conditioned medium; CMV, cytomegalovirus; eIF2α, translation initiation factor 2; ER, endoplasmic reticulum; ERAI, ER stress-activated indicator; GRP78, 78-kDa glucose-regulated protein; IIS, proinflammatory/immune suppressive; IL, interleukin 6; IRE1α, inositol-requiring enzyme 1; ISRIB, integrated stress response inhibitor; LPS, lipopolysaccharides; LysM-Cre, B6.129P2-Lys2tm1(cre)Ifo/J; MDSC, myeloid-derived suppressor cell; MFI, mean fluorescent intensity; MHC, major histocompatibility complex; miRNA, microRNA; OLS, ordinary least squares; PD-L1, programmed death ligand 1; PERK, PKR-like ER kinase; RIDD, regulated IRE1α-dependent decay; RNASeq, RNA sequencing; RT-qPCR, reverse transcriptase quantitative PCR; SERCA, sarco/endoplasmic reticulum $Ca^{2+}$-ATPase; Tapbp, tapasin; TCGA, The Cancer Genome Atlas; TERS CM, transmissible ER stress conditioned medium; Tg, thapsigargin; TGFβ, transforming growth factor β; TME, tumor microenvironment; TNFα, tumor necrosis factor α; TPM, transcripts per million; UPR, unfolded protein response; VEGF, vascular endothelial growth factor; XBP1, X-box binding protein 1; XBP-1s, spliced XBP1.

on tumor types and display phenotypic and functional diversity [2,3]. Among them, macrophages and dendritic cells—cells privileged with antigen presentation/T-cell activation functions—often acquire a mixed proinflammatory/immune-suppressive (IIS) phenotype, both in the mouse [4,5] and in humans [6,7]. Because this phenomenon is considered at the root of the dysregulation of local adaptive T-cell immunity [8,9], much emphasis has been placed on identifying common mechanisms driving the acquisition of tumor-promoting properties by macrophages and dendritic cells in the TME [5,10–14].

The TME is home to environmental *noxae* such as hypoxia and nutrient deprivation [15]. In addition, about 20% of tumors have a viral origin [16], and most (90%) solid tumors carry chromosomal abnormalities [17]. These events, independently or collectively, can lead to a dysregulation of protein synthesis, folding, and secretion [18,19] and the accumulation of misfolded proteins within the endoplasmic reticulum (ER), triggering a stress response termed the unfolded protein response (UPR) [20]. The UPR, an evolutionarily conserved adaptive mechanism [21], is mediated by three initiator/sensor ER transmembrane molecules: inositol-requiring enzyme 1 (IRE1α), PKR-like ER kinase (PERK), and activating transcription factor 6 (ATF6). In the unstressed state, these three sensors are maintained inactive through association with the 78-kDa glucose-regulated protein (GRP78) [22]. During ER stress, GRP78 disassociates from each of the three sensors to preferentially bind un-/misfolded proteins, activating each sensor and their downstream signaling cascades, which aim to normalize protein folding and secretion. PERK, a kinase, phosphorylates the translation initiation factor 2 (eIF2α) that effectively inhibits translation of most mRNAs, ultimately reducing ER client proteins. IRE1α, also a kinase, autophosphorylates and activates its RNase domain, resulting in the cleavage of the X-box binding protein 1 (XBP1) mRNA, yielding the production of the potent spliced XBP1 transcription factor isoform (XBP-1s), which drives the production of various ER chaperones to restore ER homeostasis. XBP-1s also binds to the promoter of several proinflammatory cytokine genes [23]. In addition, under ER stress or enforced autophosphorylation, IRE1α RNase domain can initiate an endonucleolytic decay of many ER-localized mRNAs, a phenomenon termed regulated IRE1α-dependent decay (RIDD) [24]. ATF6, a transcription factor, translocates to the Golgi, where it is cleaved into its functional form, and acts in parallel with XBP-1s to restore ER homeostasis [25]. If ER stress persists despite these compensatory mechanisms, the transcription factor 4 (ATF4) downstream of eIF2α activates the transcription factor CCAAT-enhancer-binding protein homologous protein (CHOP) to initiate apoptosis [20].

Although the UPR serves essentially as a cell-autonomous process to restore proteostasis, it can also act in a cell-nonautonomous way through the release of soluble molecules, a phenomenon likely to occur when cancer cells undergo an acute or unresolvable UPR [26,27]. Stress signals emanating from the ER of ER stressed cancer cells may thus result in the induction of ER stress in neighboring cells, including macrophages and dendritic cells [26,28]. This sets in motion a broad range of adaptive responses creating a functional cooperation or community effect among cells in the TME [29–31]. Under controlled experimental conditions, bone marrow–derived macrophages (BMDMs) and bone marrow–derived dendritic cells (BMDCs) cultured in conditioned media (CM) of ER stressed cancer cells develop a de novo UPR and acquire a mixed IIS phenotype [26,28] characterized by the transcriptional up-regulation of the tumorigenic proinflammatory cytokines IL-6, tumor necrosis factor α (TNFα), and IL-23 [32–34], and contextually of the immune-suppressive enzyme Arginase1 (Arg1) [35]. Under these conditions, cross-priming of naïve CD8$^+$ T cells by BMDC is greatly compromised [28]. In line with this observation, Cubillos-Ruiz reported that the incubation of BMDC in ovarian cancer CM results in *Xbp1* splicing and that the conditional knock-out (CKO) of *Xbp1* in dendritic cells improves antigen presentation and significantly reduces tumor growth in vivo [36].

In line with these observations is a report showing that GRP78 in cancer cells regulates macrophage recruitment to mammary tumors through metabolites secreted from cancer epithelial cells [37]. Thus, UPR-driven cell-nonautonomous mechanisms play a hitherto unappreciated role in orchestrating immune cells in the TME and driving their dysregulation, thus setting the stage for failure of local immune surveillance.

We therefore decided to elucidate the mechanism(s) through which the UPR may ultimately affect immune cells and perturb the TME to promote tumor growth. We focused on macrophages, as these cells represent the major population infiltrating most solid tumors in humans, conspicuously more abundant than dendritic cells and other cells of myeloid origin [38]. Relative to dendritic cells or myeloid-derived suppressor cells (MDSCs) [39,40], little is known about how the UPR affects macrophages during cancer development. Based on our earlier report that BMDM can be polarized to a mixed IIS phenotype via a UPR-mediated cell-nonautonomous mechanism [26], our initial goal was to verify whether this phenomenon could be recapitulated in tumor-infiltrating macrophages in vivo in immunocompetent mice and what UPR pathway might contribute to their dysregulation. To this day, these questions have remained largely unanswered. Here we show that the UPR and the IRE1α/XBP1 axis are activated in macrophages during tumor growth and that the CKO of IRE1α in macrophages regulates the acquisition of a mixed IIS phenotype and is also sufficient to restrain tumor development in vivo. Importantly, we discovered that IRE1α signaling regulates programmed death ligand 1 (PD-L1) expression in murine and in tumor-infiltrating macrophages in humans.

## Results

### Tumor-infiltrating CD11b$^+$ myeloid cells display the UPR/IIS signature in vivo

Previous in vitro studies indicated that BMDC and BMDM respond to a cell-nonautonomous UPR developing a complex phenotype characterized by a UPR activation and a mixed IIS phenotype [26,28]. Here, as an initial step, we interrogated tumor-infiltrating myeloid cells (CD11b$^+$) to document these characteristics during tumor growth in vivo. To this end, we implanted B16.F10 murine melanoma cells into C57BL/6 mice that carry the *Xbp1-Venus* fusion transgene under the control of the cytomegalovirus (CMV)-β actin promoter, known as the ER stress-activated indicator (ERAI) [41], which reports IRE1α-mediated *XBP1* splicing through the expression of the fluorescent Venus protein. First, we interrogated the relative abundance of CD11b$^+$ cell infiltrate into tumors 3 weeks after implantation of B16.F10 tumor cells and found that 2%–5% of the bulk tumor consisted of CD11b$^+$ myeloid cells (S1 Fig). Of these, approximately 50% expressed the F4/80 surface marker specific of macrophages. We then compared the expression of the *Venus* protein in tumor-infiltrating CD11b$^+$ cells to those in the spleen and bone marrow, both from tumor-distal and tumor-proximal femurs (Fig 1A). The *Venus* protein signal was significantly higher in tumor-infiltrating CD11b$^+$ cells relative to those in control tissues, suggesting a concurrent UPR signaling with *XBP1* splicing in the TME only.

Having established that *XBP1* splicing occurs in tumor-infiltrating CD11b$^+$ cells, we sought to detect other features of the IIS phenotype. To this end, we implanted B16.F10 cells in wild-type C57BL/6 mice and isolated by positive selection CD11b$^+$ cells from tumor, spleen, and bone marrow 22 days postimplantation. Phenotypically, the isolated cells were CD11b$^+$ and Gr1$^-$ and showed the transcriptional up-regulation of three key UPR genes: *Grp78*, a downstream target of the ATF6 pathway; *Xbp-1s*, a downstream product of the IRE1α pathway; and *Chop*, a downstream product of the PERK pathway (Fig 1B). A transcriptional up-regulation

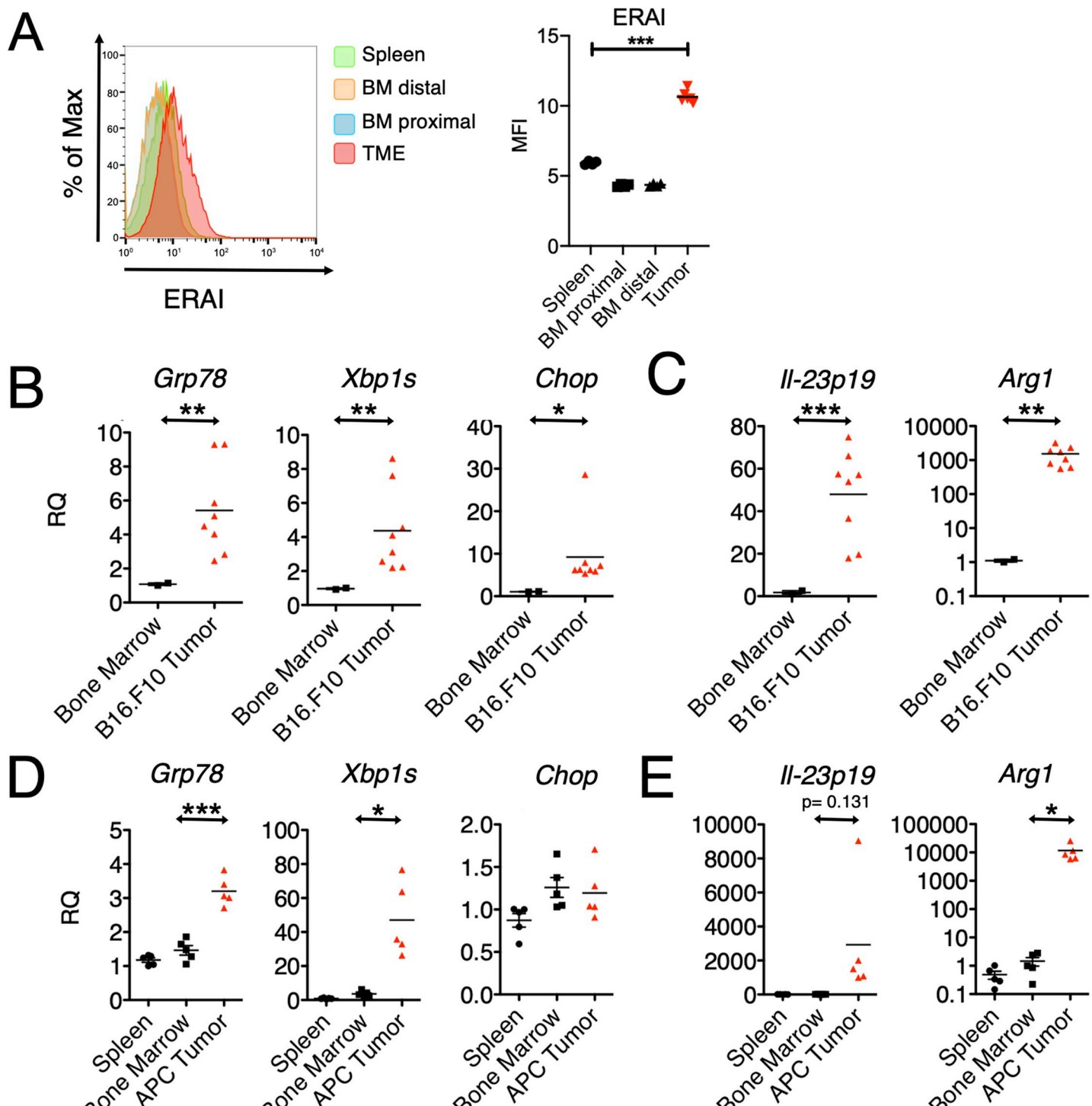

**Fig 1. Activation of the UPR and acquisition of the IIS phenotype by tumor-infiltrating CD11b+ cells in vivo.** (A) Flow cytometry histogram and comparative MFI values ($n$ = 4) of ERAI expression in CD11b+ cells resident in specified tissue. (B,C) Gene expression in CD11b+ cells isolated from B16.F10 tumors grown in C57BL/6 mice and respective bone marrow ($n \geq 2$ per group). Gene expression was arbitrarily normalized to one bone marrow sample, and values represent RQ fold transcription expression. (D,E) Gene expression in CD11b+ cells isolated from APC adenomas and respective bone marrow and spleen ($n \geq 2$ per group). RNA extracted from these cells was analyzed by RT-qPCR using specific primers. Data are included in S1 Data. APC, adenomatous polyposis coli; *Arg1*, Arginase1; BM, bone marrow; *Chop*, CCAAT-enhancer-binding protein homologous protein; ERAI, endoplasmic reticulum stress-activated indicator; *Grp78*, 78-kDa glucose-regulated protein; IIS, proinflammatory/immune-suppressive; IL, interleukin; MFI, mean fluorescent intensity; RQ, relative quantification; RT-qPCR, reverse transcriptase quantitative PCR; TME, tumor microenvironment; UPR, unfolded protein response; *Xbp1s*, spliced X-box binding protein 1.

of all three genes suggested the activation of a classical UPR. Contextually, CD11b[+] cells also showed the transcriptional up-regulation of *Il23p19*, a key proinflammatory cytokine gene, and *Arg1*, an immune-suppressive enzyme (Fig 1C).

To see if the UPR/IIS signature also hallmarks CD11b[+] cells during spontaneous tumor growth, we interrogated mice with mutations in the adenomatous polyposis coli (*Apc*) gene ("Apc mice"), which develop small-intestinal adenomas by 30 days of age [42]. We pooled CD11b[+] cell infiltrates from adenomas from multiple Apc mice and probed the expression of UPR genes *Il-23p19* and *Arg1* relative to CD11b[+] cells isolated from either the bone marrow or the spleen as controls. CD11b[+] cells from APC adenomas had increased expression of UPR genes *Il-23p19* and *Arg1* (Fig 1D and 1E). Collectively, these data suggest that CD11b[+] cells infiltrating the TME undergo ER stress and are polarized to the IIS phenotype.

## IRE1α-dependent cell-nonautonomous polarization of macrophages

Environmental conditions shown to have tumor-promoting effects have been linked to both IRE1α and PERK, making it necessary to determine which of the two was responsible for the acquisition of the IIS phenotype in our model system. To probe the role of IRE1α, we used the small molecule 4μ8C, an inhibitor specific for the RNase domain. This small molecule forms an unusually unstable Schiff base at lysine 907 (K907) and inhibits both XBP1 splicing and RIDD, but not IRE1α kinase activity. To confirm that 4μ8c (30 μM) was effective, we measured *Xbp-1* splicing in C57Bl/6 BMDM and B16.F10 cells treated with the CM of ER stressed cancer cells (transmissible ER stress CM [TERS CM]) (S2 Fig). Compared to uninhibited conditions, 4μ8C did not significantly affect the transcriptional of UPR genes (*Grp78* and *Chop*, Fig 2A). However, it significantly inhibited the transcriptional activation of *Il-6* and *Il-23p19* (Fig 2B) and trended toward inhibiting *Arg1* (*p* = 0.127) (Fig 2C). Previously, we showed that TERS CM promotes the expression of CD86 and PD-L1 in BMDC [28]. Herein, we determined that ERAI BMDMs treated with TERS CM also up-regulate CD86 and PD-L1 and that such an up-regulation that is markedly inhibited by 4μ8C (Fig 2D).

The involvement of the PERK pathway on the acquisition of the IIS phenotype by BMDM was assessed using the small molecule GSK2656157, a preferential PERK inhibitor [43]. GSK2656157 efficiently inhibited PERK phosphorylation (S3A Fig) but had no effect on the up-regulation of *Grp78*, *Il-6*, and *Arg1* induced in BMDM cultures by TERS CM (S3B Fig). Congruently, PERK inhibition had little to no effect on the surface expression of CD86 and PD-L1 (S3C Fig). Collectively, these results suggest that BMDM polarization to the IIS phenotype is IRE1α dependent.

The role of IRE1α during macrophage activation by stimuli not obviously related to the UPR was tested in experiments in which BMDMs were activated by lipopolysaccharides (LPS), a canonical activator of macrophages, or two metabolites shown to be relevant to the function of myeloid cells in the TME: lactic acid [10] and 4-hydroxynonenal (4HNE), a products of lipid peroxidation [36]. Although none of these molecules induced the transcriptional activation of *Grp78*, LPS consistently and readily induced *Il23p19* and *Il6* independent of IRE1α. Lactic acid induced *Arg1* only, and 4HNE had no effect on any of the target genes studied. Interestingly, 4μ8C reduced the induction of *Arg1* by both LPS and lactic acid, suggesting that the IRE1α may regulate the expression of this immune-suppressive molecule outside the context of the UPR (S4 Fig).

## Loss of IRE1α–Xbp1 in macrophages attenuates the IIS phenotype, PD-L1 expression, and tumor growth in vivo

Earlier reports showed that XBP1 is required for the development and survival of BMDC [44] and that the deletion of XBP1 in lymphoid dendritic cells [40,45] or in tumor-associated

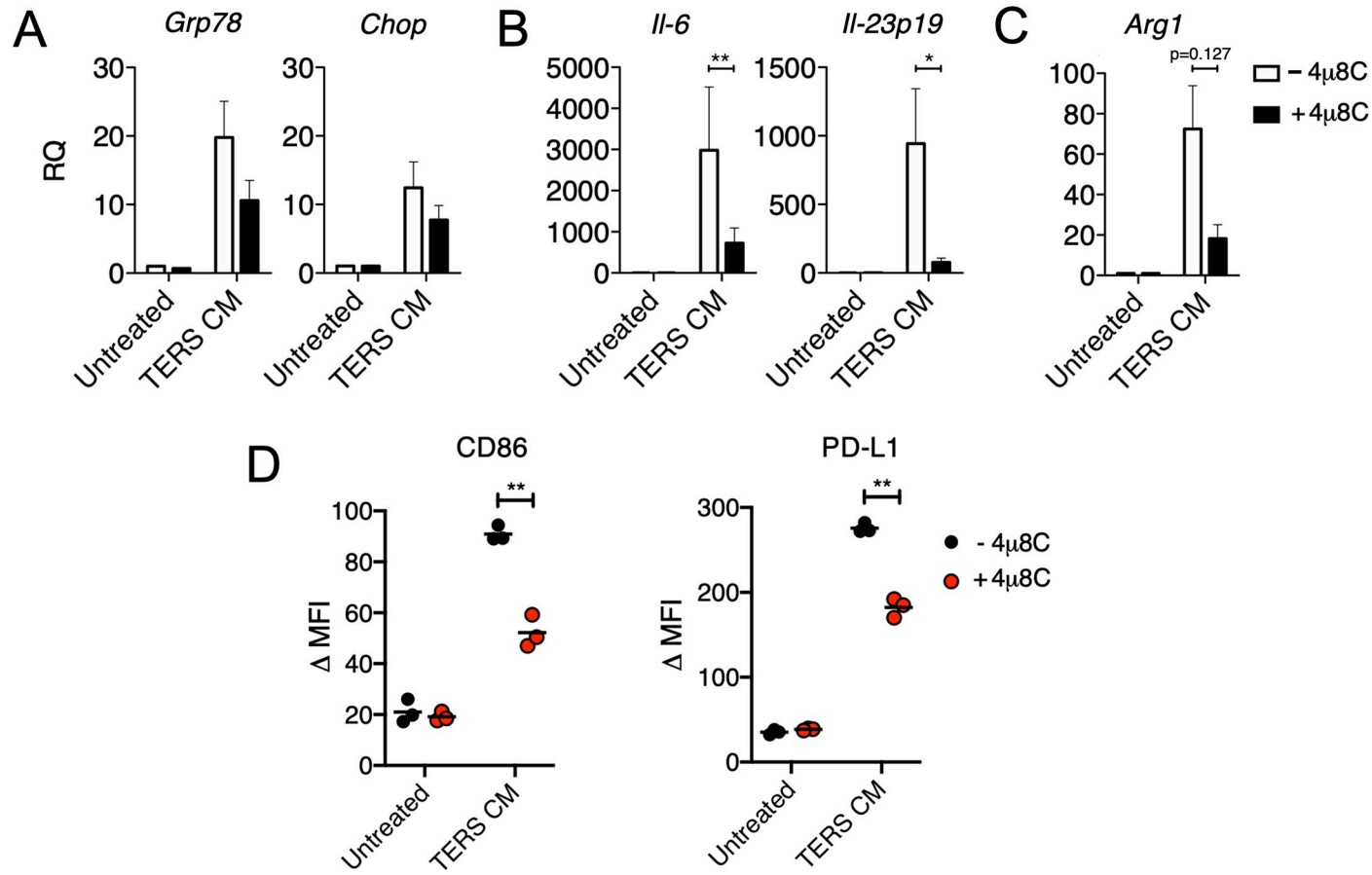

**Fig 2. Chemical IRE1α inhibition prevents IIS polarization of BMDM in vitro.** BMDM were culture in vitro in TERS CM for 18 hours with or without 4u8C (30 μM) and their mRNA subsequently tested by RT-qPCR to detect the expression of (A) UPR genes (Grp78 and Chop), (B) proinflammatory cytokines (*Il6* and *Il23p19*), and (C) immune suppression genes (*Arg1*) (*n* = 3–5 per group). RQ was determined by arbitrarily normalizing gene expression to a vehicle CM condition. Data points are expressed as means ± SEM. (D) Flow cytometry analysis of the intracellular expression of Venus protein (ERAI), and CD86 and PD-L1 surface expression in BMDM treated with TERS CM with or without 4u8C (30 μM). Data are included in S1 Data. *Arg1*, Arginase1; BMDM, bone marrow–derived macrophage; *Chop*, CCAAT-enhancer-binding protein homologous protein; CM, conditioned medium; ERAI, endoplasmic reticulum stress-activated indicator; *Grp78*, 78-kDa glucose-regulated protein; IIS, proinflammatory/immune-suppressive; *Il*, interleukin; IRE1α, inositol-requiring enzyme 1; MFI, mean fluorescent intensity; PD-L1, programmed death-ligand 1; RQ, relative quantification; RT-qPCR, reverse transcriptase quantitative PCR; TERS CM, transmissible ER stress CM; UPR, unfolded protein response.

dendritic cells [36] improves antigen cross-priming and reduces tumor (ovarian) growth in the mouse. The role of the IRE1α/XBP1 axis in macrophage activation in the context of tumorigenesis has not been previously explored. Chemical inhibition of IRE1α endonuclease activity clearly implicated the IRE1α pathway in macrophage polarization to the IIS phenotype. However, since 4μ8C inhibits both *Xbp1* splicing and RIDD activity [46], we used a genetic approach to distinguish mechanistically among the two IRE1α functions in the acquisition of the IIS phenotype. To this end, we developed mice with *Ern1* (the gene coding for IRE1α) or *Xbp1* CKO in macrophages by breeding mice floxed (*fl/fl*) for *Ern1* [41] or *Xbp1* [47] with B6.129P2-Lys2tm1(cre)Ifo/J (LysM-Cre) mice [48]. The genotype of CKO mice is shown in S5 Fig. Western blot analysis of *Ern1*-CKO BMDM confirmed the absence of IRE1α (Fig 3A) as well as the absence of the spliced form of *Xbp1* following treatment with the sarco/endoplasmic reticulum Ca²⁺-ATPase (SERCA) inhibitor thapsigargin (Tg) (Fig 3B). Under similar experimental conditions, *Xbp1*-CKO BMDM showed an intact IRE1α expression under basal conditions (Fig 3A) but the absence of the spliced form of *Xbp1* after Tg treatment (Fig 3C). Thus,

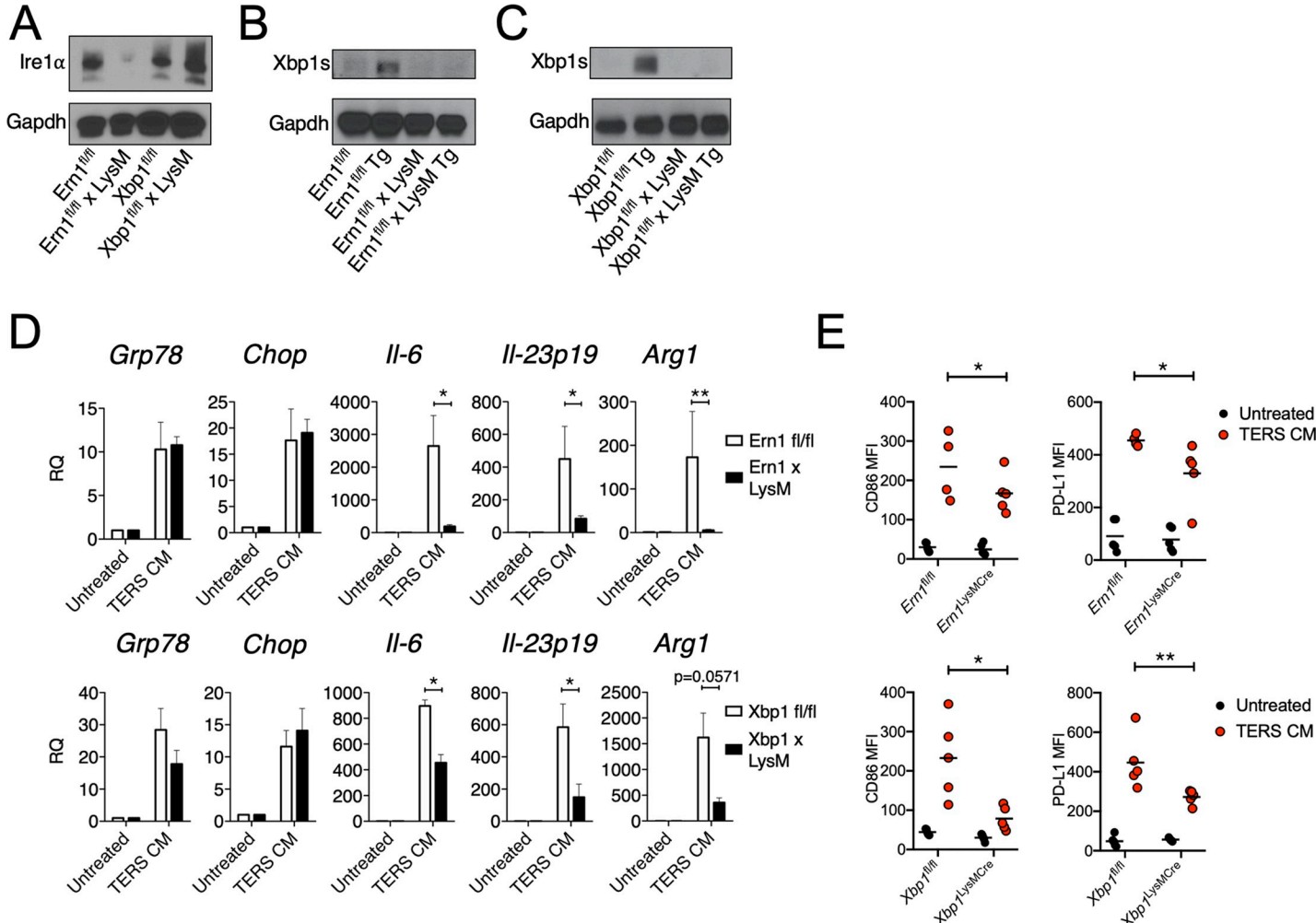

**Fig 3. Deficiency in the IRE1α-XBP1 axis in macrophages attenuates the IIS phenotype, PD-L1 expression, and tumor growth.** (A) Western blot analysis of *Ern1*-CKO BMDM showing lack of Ire1α upon activation (24 hours) by Tg (300 nM). (B) Western blot analysis of *Ern1*-CKO BMDM showing lack of Xbp1s following activation (24 hours) by Tg (300 nM). (C) Western blot analysis of *Xbp1*-CKO BMDM showing lack of Xbp1s following activation (24 hours) by Tg (300 nM). (D) RT-qPCR analysis of UPR and IIS genes in wild-type or CKO BMDM untreated or treated with TERS CM. Values represent the mean ± SEM (*n* = 3–5 per group). (E) IRE1α-XBP1 deficiency reduces CD86 and PD-L1 expression in BMDM. *Ern1*fl/fl, *Xbp1*fl/fl, *Ern1*-CKO, and *Xbp1*-CKO BMDM were treated (18 hours) with TERS CM and subsequently stained with PE-conjugated antibodies to CD86 and CD274. The MFI for both surface proteins was quantified and plotted against the MFI of the corresponding unstimulated control. Statistical significance was determined using the Mann-Whitney *t* test. (*n* = 4–5 mice per group). Data are included in S1 Data. *Arg1*, Arginase1; BMDM, bone marrow–derived macrophage; *Chop*, CCAAT-enhancer-binding protein homologous protein; CKO, conditional knock-out; Gapdh, glyceraldehyde 3-phosphate dehydrogenase; *Grp78*, 78-kDa glucose-regulated protein; IIS, proinflammatory/immune-suppressive; *Il*, interleukin; IRE1α, inositol-requiring enzyme 1; MFI, mean fluorescent intensity; PD-L1, programmed death ligand 1; PE, phycoerythrin; RQ, relative quantification; RT-qPCR, reverse transcriptase quantitative PCR; TERS CM, transmissible ER stress conditioned medium; Tg, thapsigargin; UPR, unfolded protein response; XBP1, X-box binding protein 1; Xbp1s, spliced Xbp1.

the LysM-Cre CKO system was effective at specifically deleting IRE1α and Xbp1 in activated BMDM.

First, we compared the transcriptional response of *Ern1*- and *Xbp1*-CKO vs wild-type BMDM when treated with TERS CM. We found that *Grp78* and *Chop* were unaffected in *Ern1*-CKO BMDM, but *Il6*, *Il-23p19*, and *Arg1* were markedly and significantly reduced in CKO relative to *fl/fl* control BMDM (Fig 3D, upper panels). Likewise, in *Xbp1*-CKO BMDM, the induction of *Grp78* and *Chop* was unaffected, but the activation of *Il6* and *Il23p19* was significantly reduced compared to *fl/fl* control BMDM. The activation of *Arg1* trended lower in

*Xbp1*-CKO compared to *fl/fl* control BMDM ($p$ = 0.0571) (Fig 3D, lower panels). These results confirm that the IRE1α-XBP1 axis mediates the IIS phenotype.

We then evaluated the effect of TERS CM on the expression of CD86 and PD-L1 in BMDM populations. In vitro treatment of *Ern1*- or *Xbp1*-CKO BMDM with TERS CM yielded a significant reduction of both surface proteins compared to wild-type BMDM (Fig 3E). Thus, the conditional deletion of the IRE1α/XBP1 axis in macrophages produced effects consistent with the pharmacological inhibition by 4μ8C. This suggests that the IRE1α-XBP1 axis is central to both macrophage activation (CD86 up-regulation) and the acquisition of PD-L1, a marker of immune disfunction. We ruled out the possibility that PD-L1 expression was the result of canonical IFN-γ signaling because (1) we did not detect IFN-γ in TERS CM (S6A Fig), (2) a blocking antibody to human IFN-γ had no effect on *Cd274* gene expression in BMDM treated with TERS CM (S6B Fig), and (3) RNA sequencing (RNASeq) data showed no induction of the *Ifng* gene in either *Ern1*-CKO or *fl/fl* control BMDM treated with TERS CM (S6C Fig).

To ascertain the physiological relevance of these findings, we next assessed the survival of *Ern1*- and *Xbp1*-CKO mice implanted with B16.F10 melanoma cells. We reasoned that survival would constitute an optimal initial readout for the complex interactions between cancer cells and immune cells in the TME with focus on the IRE1α-XBP1 axis in myeloid cells. Survival in *Ern1*-CKO mice was significantly greater ($p$ = 0.03) than in control *Ern1 fl/fl* mice (Fig 4A). By contrast, *Xbp1*-CKO mice survived longer than control *Xbp1 fl/fl* mice, but the difference was nonsignificant (Fig 4A). Based on survival data, we isolated F4/80 tumor-infiltrating macrophages of tumor-bearing *Ern1*-CKO mice to assess the UPR/IIS and *Cd274* gene expression status. *Xbp1s*, *Il-23p19*, *Arg1*, and *Cd274* genes were all markedly reduced in *Ern1*-CKO macrophages compared to their *Ern1 fl/fl* counterpart (Fig 4B). Together, these results point to macrophage IRE1α as a key negative regulator of TME immunodynamics and tumor growth in vivo.

## Loss of RIDD regulation in *Ern1*-CKO macrophages

Because the IRE1α-XBP1 axis also regulates PD-L1 expression and both *Ern1*- and *Xbp1*-CKO BMDM showed significantly reduced surface PD-L1 protein expression compared to *fl/fl* BMDM (Fig 3E), we decided to distinguish the relative contribution of *Xbp1* splicing and RIDD to this phenomenon. To this end, we performed reverse transcriptase quantitative PCR (RT-qPCR) on *Ern1*- and *Xbp1*-CKO BMDM treated or not with TERS CM relative to *fl/fl* controls. We found that *Cd274* gene transcription was markedly and significantly lower in *Ern1*-CKO BMDM relative to *fl/fl* controls (Fig 5A). By contrast, *Xbp1*-CKO BMDM and *fl/fl* BMDM had comparable *Cd274* gene transcription values (Fig 5A). Based on this result and on PD-L1 surface expression (Fig 3E), we tentatively conclude that XBP1-mediated regulation of PD-L1 occurs at the posttranslational level, whereas IRE1α-mediated regulation is a transcriptional event. This conclusion favors the view that IRE1α-mediated PD-L1 regulation may occur via RIDD, justifying an in-depth analysis of RIDD activity in *Ern1*-CKO BMDM.

We performed RNASeq analysis of *fl/fl* and *Ern1*-CKO BMDM untreated or treated with TERS CM. Three independently derived BMDM populations per group were analyzed. The genotype of each mouse used in this experiment is shown in S5 Fig. Upon TERS CM treatment, *Ern1* expression in *Ern1*-CKO macrophages was 1.79-fold over that of untreated cells compared to 3.26-fold in *fl/fl* macrophages (Fig 5B). We found that consistent with the flow cytometry data, *Cd274* (PD-L1) expression was markedly increased in macrophages (44.45-fold) but only moderately increased in *Ern1*-CKO macrophages (4.11-fold, Fig 5C). Thus, both genetic and chemical inhibition of IRE1α signaling yielded concordant results.

Next, we performed a comprehensive analysis of RIDD activity using a set of 33 putative RIDD target genes previously defined [49]. We found that only half (16) of these genes

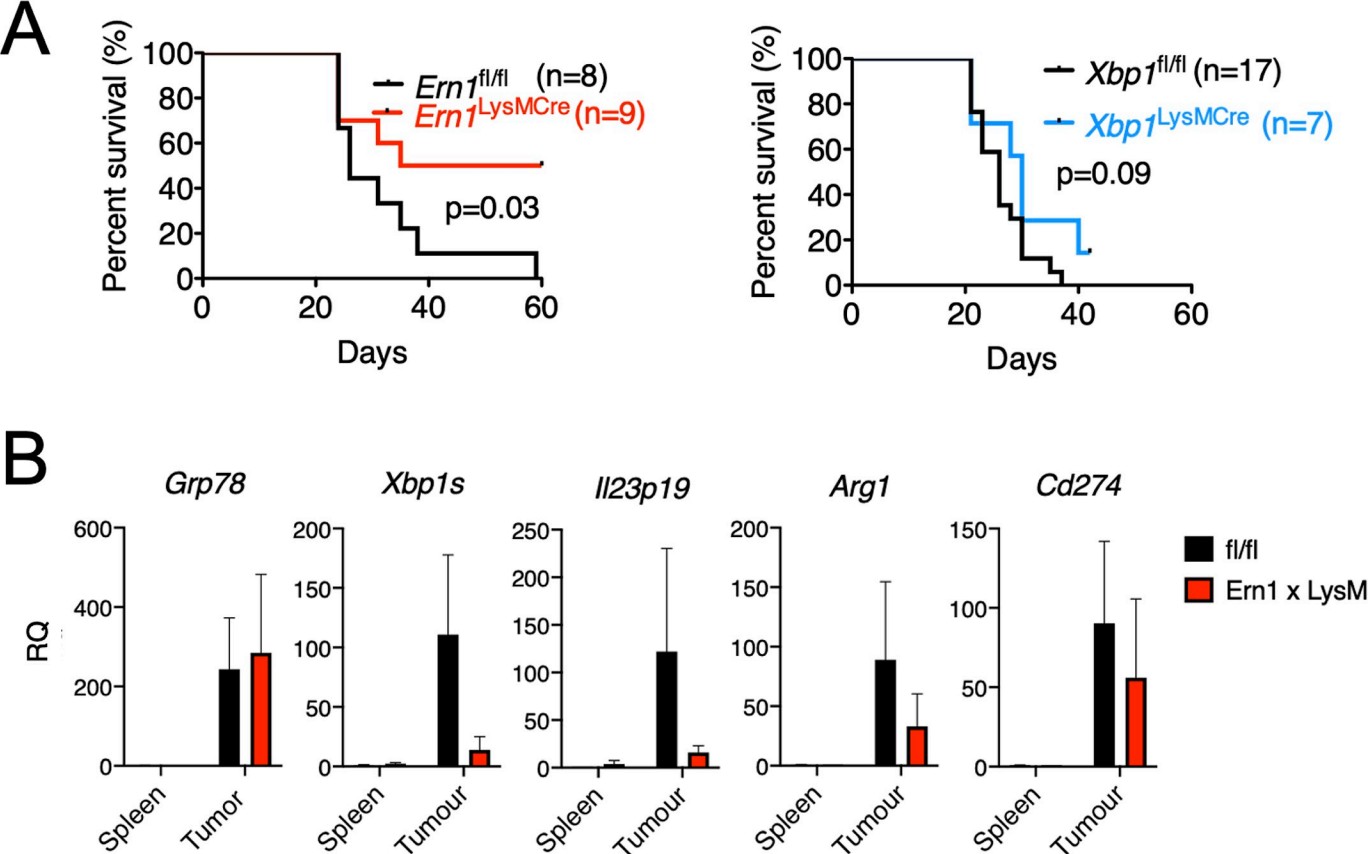

**Fig 4. Tumor growth and tumor-infiltrating macrophage analysis in *Ern1/Xbp1*-CKO mice.** (A) Kaplan-Meier survival curves of *Ern1fl/fl*, *Xbp1fl/fl*, *Ern1*-CKO, and *Xbp1*-CKO mice injected in the right flank with 3x10e4 B16.ERAI cells/mouse. Tumor measurements were taken every 2 days in two dimensions. Mice were euthanized once tumors reached 20 mm in either dimension. (B) Gene expression in F4/80+ macrophages isolated from B16.F10 tumors implanted in *Ern1*-CKO or *fl/fl* mice, and respective spleen controls (*n* = 2 per group). mRNA was extracted enzymatically using the Zygem RNAgem Tissue PLUS kit. Gene expression was arbitrarily normalized to one spleen sample and values represent RQ fold transcript expression. Data points are expressed as means ± SEM. Data are included in S1 Data. *Arg1*, Arginase1; CKO, conditional knock-out; ERAI, ER stress-activated indicator; *Grp78*, 78-kDa glucose-regulated protein; *Il*, interleukin; PD-L1, programmed death ligand 1;; RQ, relative quantification; *Xbp1*, X-box binding protein 1; *Xbp1s*, spliced Xbp1.

behaved as bona fide RIDD targets in TERS CM–treated BMDM (i.e., decreased expression after TERS CM treatment in *fl/fl* macrophages) (Fig 5D, upper panel). We found that in *Ern1*-CKO macrophages, there was a clear loss of a "RIDD signature" compared to *fl/fl* macrophages, both basally and after TERS CM treatment (Fig 5D, lower panel). When considered together through an analysis of the mean *z* score for the 16 genes, it became apparent that TERS CM induction of RIDD activity was much more effective in *fl/fl* than in *Ern1*-CKO macrophages (Fig 5E). Collectively, these results show that macrophages lacking *Ern1* lose RIDD regulation, suggesting that RIDD may be implicated in the regulation of PD-L1 expression.

In the same analysis, we found that *Tapbp* (tapasin), a chaperone molecule involved in the stabilization of high-affinity peptide/major histocompatibility complex (MHC)-I complexes in the ER [50], did not behave as RIDD. In fact, *fl/fl* macrophages treated with TERS CM showed increased, not diminished, expression at variance with previous reports on lymphoid (CD8α+) dendritic cells [40,45]. The expression of *Bloc1s1* (a canonical RIDD target) was reduced, confirming that TERS CM induces RIDD (S7A Fig). RT-qPCR analysis of *Tapbp* in *Xbp1 fl/fl* macrophages showed similar results (S7B Fig). Perhaps, *Tapbp* is regulated by RIDD differently in CD8α+ dendritic cells and in BMDM.

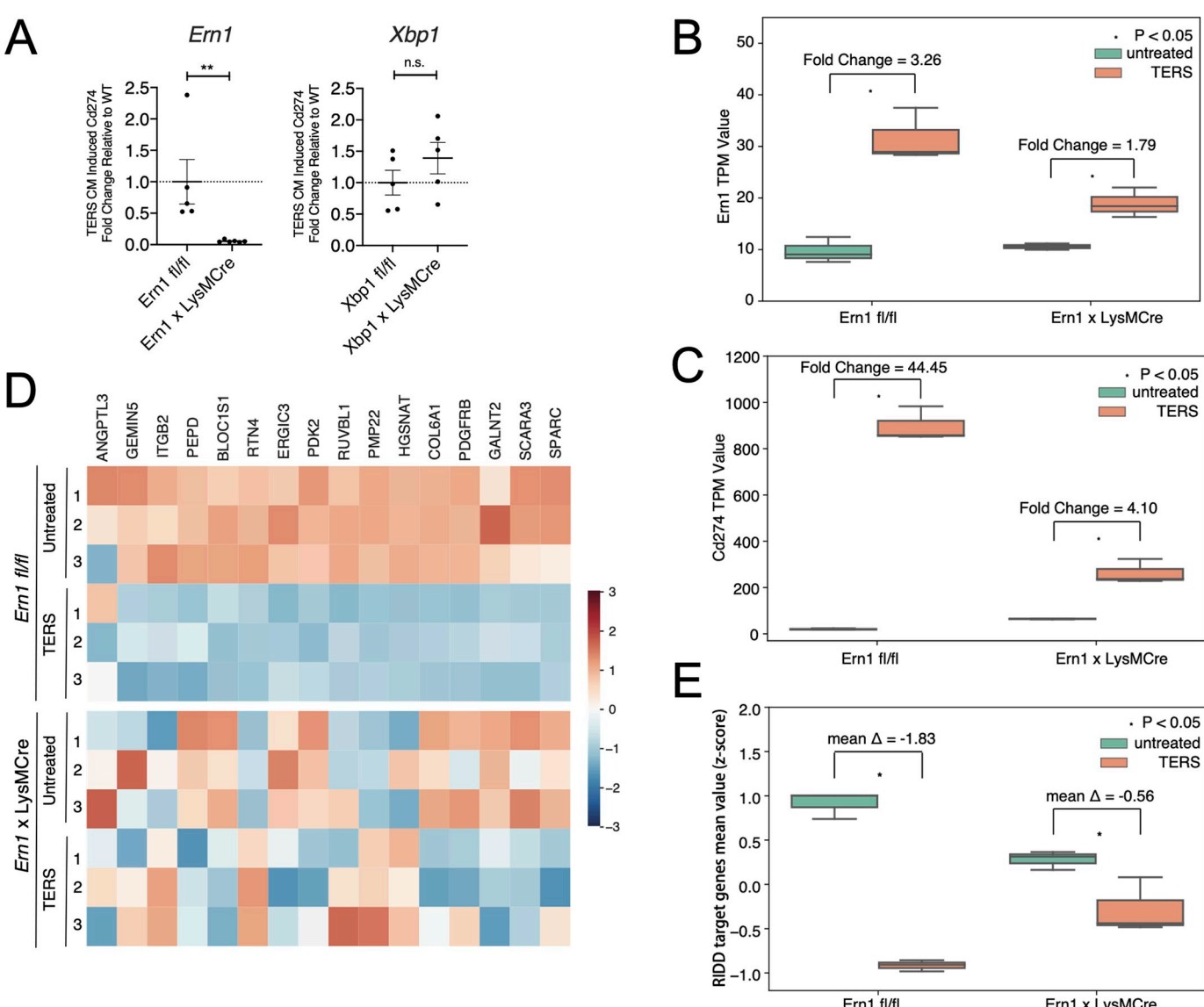

**Fig 5. RIDD analysis of WT and *Ern1*-CKO BMDM treated with TERS CM.** (A) Fold change in *Cd247* (PD-L1) transcription in *Ern1*-deficient (left panel) and *XBP1*-deficient (right panel) BMDMs activated with TERS CM. (B) RNASeq analysis of *Ern1* expression in untreated or TERS CM–treated WT or *Ern1*-CKO BMDM. TERS CM–induced fold changes are indicated in the graph. (C) Heatmap showing the relative expression of 16 RIDD target genes in untreated or TERS CM–treated WT or *Ern1*-CKO BMDM. (D) RNASeq analysis of *Cd274* expression in untreated or TERS CM–treated WT or *Ern1*-CKO BMDM. TERS CM–induced fold changes are indicated in the graph. (E) Comparison of mean *z* scores for the 16 RIDD target genes in untreated or TERS CM–treated WT or *Ern1*-CKO BMDM. RNASeq data have been deposited in BioProject database (accession no. ID PRJNA622650; http://www.ncbi.nlm.nih.gov/bioproject/622650). Data are included in S1 Data. BMDM, bone marrow–derived macrophage; CKO, conditional knock-out; CM, conditioned medium; n.s., nonsignificant; PD-L1, programmed death ligand 1; RIDD, regulated IRE1α-dependent decay; RNASeq, RNA sequencing; TERS CM, transmissible ER stress CM; TPM, transcripts per million; WT, wild type; XBP1, X-box binding protein 1.

## A link between IRE1α and PD-L1 expression in human tumor-infiltrating macrophages

The data reported herein suggest that *Cd274* gene expression in murine macrophages is positively regulated by IRE1α. Recently, Xu and colleagues [51] reported that PD-L1 protein expression in murine MYC[tg]:KRAS[G12D] tumor cells is decreased by a small molecule that enables the cell to resume translation while the eIF2α downstream from PERK remains

phosphorylated. Therefore, we decided to study the relationship between *CD274* (PD-L1) gene expression and the two major UPR pathways, IRE1α and PERK, across multiple human cancers. We began by interrogating the relative contribution of *ERN1* (IRE1α) and *EIF2AK3* (PERK) to *CD274* gene expression. In this analysis, we queried The Cancer Genome Atlas (TCGA) collection of RNASeq expression data for bulk samples from 31 tumor types. Across these data, we observed that *ERN1* correlates strongly with *EIF2AK3* (Pearson correlation coefficient = 0.55; $p < $ 1e-200) (S8A Fig) and that both *ERN1* ($p \leq$ 1.46e-51) and *EIF2AK3* ($p \leq$ 1.62e-44) correlate positively with *CD274*, suggesting that the UPR plays a role in *CD274* gene expression. These correlations prompted us to further interrogate the relationship between *CD274*, *ERN1*, and *EIF2AK3*, with respect to levels of infiltrating macrophages in bulk tumor samples approximated by a macrophage score derived from the geometric mean of three genes expressed by macrophages (*CD11b*, *CD68*, and *CD163*). We found a positive correlation between *ERN1* and *CD274* within the high macrophage infiltration group ($>$70th percentile) (Spearman correlation coefficient 0.18; $p < $ 1.3e-21) (S8B Fig). By contrast, the low macrophage infiltration group ($<$30th percentile) had a much weaker correlation (Spearman correlation coefficient 0.06; $p < $ 0.001) (S8B Fig). On the other hand, *EIF2AK3* and *CD274* within the high macrophage infiltration group ($>$70th percentile) had a lower correlation (Spearman correlation coefficient 0.09; $p < $ 1.9e-7) than in the corresponding *Ern1* group (S8C Fig). Finally, *EIF2AK3* and *CD274* within the low macrophage infiltration group ($<$30th percentile) had a surprisingly higher correlation (Spearman correlation coefficient 0.15; $p < $ 8.32e-15) than in the respective high macrophage infiltration group (S8C Fig). Collectively, this analysis suggests that when macrophage infiltration is high, *Ern1* is a better predictor of *CD274* gene expression than *EIF2AK3*.

We also integrated the macrophage score with *ERN1* and *EIF2AK3* to predict *CD274* expression in an ordinary least squares (OLS) linear regression model, including the tumor type as a covariate (Table 1). We found that this model assigns significant, positive coefficients for the interaction terms of macrophages with *ERN1* (*ERN1*\**Macrophages, beta coefficient = 0.0012, $p < $ 0.023) but not *EIF2AK3* (*EIF2AK3*\**Macrophages, beta coefficient = 0.0007, $p < $ 0.155), suggesting that *ERN1* but not *EIF2AK3* is predictive of *CD274* gene expression within tumor-infiltrating macrophages in individual tumor types (Table 1).

To validate these results, we analyzed RNASeq data generated from macrophages isolated from 13 patients with either endometrial or breast cancer [52]. We found a strong Pearson correlation coefficient between *ERN1* and *EIF2AK3* in these data (correlation coefficient 0.738; $p < $ 0.003), suggesting UPR activation. Since IRE1α activity is a multistep and complex process [53] and may not be completely captured by *ERN1* expression levels, we derived a systemic representation of pathway activity controlled by IRE1α and, by comparison, PERK. We collected sets of downstream genes in the IRE1α and PERK pathways [54] and derived aggregate scores for each pathway from the mean expression signal of all detectable genes after *z* score transformation. Since the transformed pathway scores could potentially amplify noise from genes with low expression, we applied filters to include only genes in each pathway with levels beyond a specific threshold (S9 Fig). We varied this filter threshold from zero to 1,000 raw counts and then included the pathway activity scores in multiple OLS linear models to predict *CD274* across tumor-infiltrating macrophage samples (Fig 6A). We found that a filter threshold of 100 counts effectively reduced noise while preserving signal from 84% of detectable genes in both the IRE1α and PERK pathways. In this model, the IRE1α score predicted *CD274* expression with a significant positive beta coefficient (beta coefficient = 21.043, $p$-value = 0.040), whereas the PERK score was nonsignificant (beta coefficient = 36.842, $p$-value = 0.103). This pattern of significant IRE1α coefficient and nonsignificant PERK coefficient was consistent across all filter thresholds (Fig 6B). Comparing models wherein *CD274*

**Table 1. Predicted CD274 gene expression in tumor-infiltrating macrophages in different tumor types.**

| Number | Name | Coefficient | *p*-Value | Tumor Type |
|---|---|---|---|---|
| 0 | Intercept | 0.56812244 | 0.63071856 | NA |
| 1 | BLCA | 3.73078163 | 0.00291006 | Bladder urothelial carcinoma |
| 2 | BRCA | −0.2811479 | 0.81470058 | Breast invasive carcinoma |
| 3 | CESC | 6.89337911 | 8.34E-08 | Cervical squamous cell carcinoma |
| 4 | CHOL | 0.40146841 | 0.84136569 | Cholangiocarcinoma |
| 5 | COAD | 0.53007019 | 0.66721643 | Colon adenocarcinoma |
| 6 | DLBC | 20.8423568 | 8.08E-29 | Large B cell lymphoma |
| 7 | ESCA | 3.01786009 | 0.03778129 | Esophageal carcinoma |
| 8 | GBM | −0.5684441 | 0.68831792 | Glioblastoma multiforme |
| 9 | HNSC | 5.61635157 | 5.51E-06 | Head and neck squamous cell carcinoma |
| 10 | KICH | 5.1784745 | 0.00237693 | Kidney chromophobe |
| 11 | KIRC | −0.3455225 | 0.77961985 | Kidney renal clear cell carcinoma |
| 12 | KIRP | 0.66598396 | 0.60609618 | Kidney renal papillary cell carcinoma |
| 13 | LGG | −0.6396981 | 0.60711394 | Brain lower grade glioma |
| 14 | LIHC | −0.2908561 | 0.81603644 | Liver hepatocellular carcinoma |
| 15 | LUAD | 3.69555473 | 0.00272697 | Lung adenocarcinoma |
| 16 | LUSC | 6.86545798 | 3.49E-08 | Lung squamous cell carcinoma |
| 17 | MESO | 1.14121912 | 0.46922913 | Mesothelioma |
| 18 | OV | −0.159162 | 0.90070016 | Ovarian serous cystadenocarcinoma |
| 19 | PAAD | −0.7801816 | 0.56830669 | Pancreatic adenocarcinoma |
| 20 | PCPG | 2.24302929 | 0.09976162 | Pheochromocytoma and paraganglioma |
| 21 | PRAD | −0.1264243 | 0.91904749 | Prostate adenocarcinoma |
| 22 | READ | 0.32928907 | 0.81240957 | Rectum adenocarcinoma |
| 23 | SARC | −1.0905281 | 0.40928964 | Sarcoma |
| 24 | SKCM | 0.86168428 | 0.57767244 | Skin cutaneous melanoma |
| 25 | STAD | 4.07073499 | 0.0014699 | Stomach adenocarcinoma |
| 26 | TGCT | 0.87213696 | 0.54559414 | Testicular germ cell tumors |
| 27 | THCA | 2.3173613 | 0.06056962 | Thyroid carcinoma |
| 28 | THYM | 13.0435304 | 4.96E-19 | Thymoma |
| 29 | UCEC | −0.0479321 | 0.96878827 | Uterine corpus endometrial carcinoma |
| 30 | UCS | −0.5462166 | 0.75460409 | Uterine carcinosarcoma |
| 31 | UVM | 0.12508091 | 0.9379339 | Uveal melanoma |
| 32 | ERN1 | −0.0263998 | 0.34478081 | NA |
| 33 | GmeanMacro | 0.0425603 | 1.93E-07 | NA |
| 34 | ERN1:gmeanMacro | 0.00121149 | 0.02397205 | NA |
| 35 | EIF2AK3 | −0.0029254 | 0.88327969 | NA |
| 36 | EIF2AK3:gmeanMacro | 0.00078209 | 0.15506699 | NA |

expression was explained by IRE1α activity alone or by both IRE1α and PERK activity using the Akaike information criterion (AIC) analysis shows that a model containing both is 0.54 times as probable as the IRE1α alone to minimize the information loss (ΔAIC = 1.23). Taken together, these analyses suggest that the activation of *CD274* gene expression in tumor-infiltrating macrophages depends primarily on the IRE1α pathway.

## Discussion

Here we analyzed the effect of the UPR on gene expression regulation in macrophages as a potential mechanism driving immune dysregulation in the TME. Tumor-infiltrating CD11b[+]

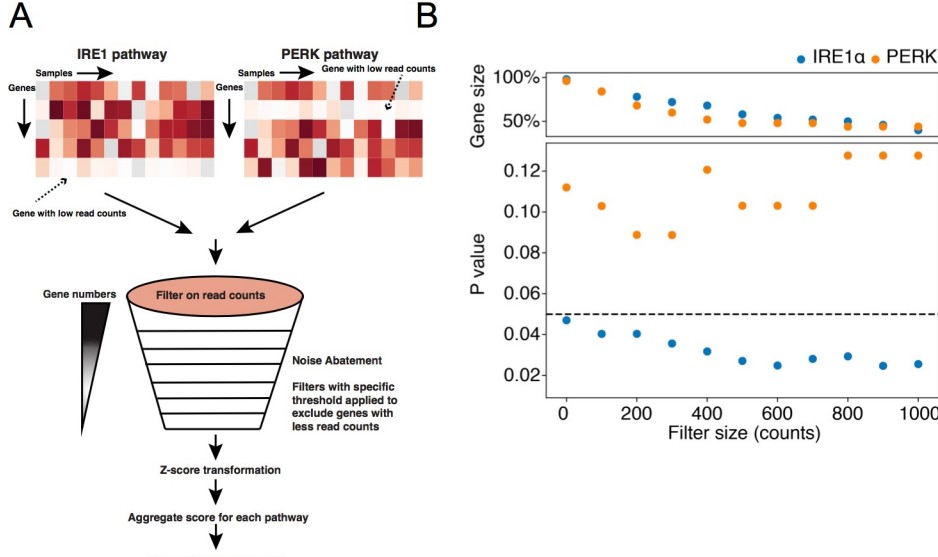

**Fig 6. OLS linear model prediction of *CD274* gene expression in human tumor-associated macrophages.** (A) An illustration of the development of aggregated pathway scores. For both pathways we used filters with different thresholds to filter out genes with less read counts to account for baseline technical artifacts. Then, we *z* score transformed both the gene matrix for both pathways and aggregated these scores to predict *CD274* gene expression. (B) RNASeq data from tumor-associated macrophages isolated from 13 human endometrial or breast cancer samples were analyzed using 11 OLS linear models for each pathway (IRE1α or PERK). Each model was applied using different filters, each representing increasing read count thresholds. In the upper panel, each dot represents the fraction of genes remaining in the model after a given filter was applied. In the lower panel, the *p*-value for each pathway predicting PD-L1 gene expression is indicated at each read count threshold. Data are included in S1 Data. IRE1α, inositol-requiring enzyme 1; OLS, ordinary least squares; PD-L1, programmed death ligand 1; PERK, PKR-like ER kinase; RNASeq, RNA sequencing.

myeloid cells in B16.F10 tumors and in spontaneously arising colonic adenomas in Apc mice have an active UPR and display a mixed IIS phenotype. Using both a pharmacologic and genetic approach, we show that the IRE1α/XBP1 axis plays a central role in macrophage activation and polarization to a mixed phenotype, including the up-regulation of PD-L1. In agreement with the mouse data, we found that in human tumor-infiltrating macrophages *CD274* (PD-L1) gene transcription correlates significantly with the IRE1α gene signature. B16.F10 tumor–bearing mice with conditional *Ern1*-KO but not *Xbp1*-KO macrophages had significantly greater survival than their *fl/fl* controls. Collectively, these results show that IRE1α signaling drives macrophage dysregulation, impacting negatively the immunobiology of the TME and ultimately the host's ability to control tumor growth.

Virtually all adult solid tumors (carcinomas most notably) contain infiltrates of diverse leukocyte subsets, including macrophages, dendritic cells, and lymphocytes [2]. CIBERSORT and immunohistochemical tools have previously shown that macrophages represent the largest fraction among infiltrating leukocytes and their density correlates directly with poor survival [38,55]. In the mouse, tumor-infiltrating CD11b[+] myeloid cells produce proinflammatory/pro-tumorigenic cytokines (IL-6, IL-23, TNFα) [32–34] but, oddly, also anti-inflammatory cytokines (IL-10, transforming growth factor β [TGFβ]) and molecules with immune-suppressive function (Arg1, peroxynitrite, and indoleamine 2–3 dioxygenase) [8]. In humans, monocytes/macrophages with a "mixed" proinflammatory/suppressive phenotype have been reported in patients with renal cell carcinoma [6] and breast cancer [7]. Thus, a dysregulation-prone TME harbors CD11b[+] myeloid cells with a split IIS phenotype that may be the result of hijacking by tumors for their own benefit [56]. Indeed, we previously proposed that tumor-derived UPR-

driven factors determine the IIS phenotype in myeloid cells [57], contributing to progressive immune dysregulation and failure of immune surveillance.

Here, we analyzed two murine tumor models to demonstrate that tumor-infiltrating CD11b[+] cells display features of UPR activation and a mixed IIS phenotype. The results clearly show that the UPR is associated with myeloid cell polarization in vivo but do not allow a distinction between a cell-autonomous and a cell-nonautonomous mechanism. However, since common triggers of inflammation, such as LPS, or TME metabolites, such as 4HNE and lactic acid [10, 36], did not induce a UPR/IIS phenotype, we favor the possibility that these changes in myeloid cells result from a cell-nonautonomous mechanism of intercellular communication consistent with findings on BMDM and BMDC analyzed under controlled in vitro conditions [26,28]. This appears to be a general mechanism because we recently showed that cell-nonautonomous intercellular communication among cancer cells induces an adaptive UPR imparting receiver cells with enhanced cellular fitness and resistance to various stressors [27].

A pharmacological approach using a small molecule (4μ8c) that inhibits IRE1α significantly reduced the transcription of *Il-6* and *Il-23p19* induced by TERS CM, demonstrating a direct involvement of the IRE1α/XBP1 axis in driving proinflammation during an adaptive UPR. This is consistent with previous reports showing that XBP1 is recruited to the *Il6* and *Il23* promoters [23] and that *Il23* transcription is IRE1α-dependent [58]. Interestingly, 4μ8C did not reduce the transcription of these cytokines in the absence of a UPR, implying that IRE1α selectively regulates proinflammation within the boundaries of the UPR. Our findings on macrophage polarization via cell-nonautonomous means are consistent with reports showing that IRE1α drives M1 to M2 polarization of macrophages within white adipose tissue [59] and their inflammatory response to saturated fatty acids [60]. Our findings are consistent with a recent report showing that in cystic fibrosis, lung macrophages undergo polarization imbalance, with suppression of M2 polarization associated with activation of the IRE1α/XBP1 axis. An overactive IRE1α/XBP1 axis was further found to reprogram macrophages toward an increased metabolic state, with increased glycolytic rates and mitochondrial function, and an exaggerated production of TNFα and IL-6 [61].

Importantly, 4μ8C also inhibited the TERS CM–induced up-regulation of *Arg1* and that of proangiogenic vascular endothelial growth factor (VEGF) (S10 Fig). Since IL-6 and IL-23 are known to bias T-cell differentiation toward inflammatory (Th17) or regulatory T cells [62–66], and Arg1 potently suppresses the clonal expansion of T cells activated by antigen [28,35], it follows that signaling through the IRE1α/XBP1 axis is of paramount importance to the economy of the TME and may be at the origin of a loss of local immune surveillance. A role of the IRE1α/XBP1 axis in the more general context of immune cell dysregulation in the TME where there exists metabolic pressure. In ovarian cancer, the IRE1α/XBP1 axis was shown to cripple T-cell metabolism and that T cells lacking XBP1 have superior antitumor immunity, delayed malignant progression, and increased overall survival [67]. Taken together, these data suggest that pharmacologic interventions to specifically mitigate the activity of the IRE1α/XBP1 axis in immune cells participating to the adaptive immune response could produce synchronous beneficial antitumor effects.

*Ern1-* or *Xbp1*-CKO macrophages enabled us to distinguish different roles within the IRE1α/XBP1 axis relative to immune dysregulation and tumor growth. In vitro, both *Ern1-* and *Xbp1*-CKO BMDM had decreased activation (CD86 and PD-L1 surface expression) and an attenuated IIS phenotype compared to control *fl/fl* macrophages when cultured in TERS CM, consistent with the effects of 4μ8C. However, only IRE1α deficiency significantly increased survival of mice implanted with B16.F10 melanoma cells, a result possibly reflected by an attenuation of the UPR/IIS signature and PD-L1 in tumor-infiltrating macrophages. Cubillos-Ruiz [36] also observed that IRE1α deficiency in dendritic cells yielded greater

survival than XBP1 deficiency in a model of ovarian cancer. By inference, we showed that B16. F10 tumor cells admixed with BMDC with a UPR/IIS phenotype form faster-growing and larger tumors that had a marked reduction in tumor-infiltrating CD8[+] T cells [28].

Chemical and genetic inhibition both showed that IRE1α regulates the surface expression of PD-L1 triggered in an IFNγ-independent manner through an adaptive UPR. PD-L1 activation is considered to occur mainly in response to IFNγ, albeit other mechanisms can contribute to its activation both at the transcriptional and posttranslational levels [68]. The inhibition of cell surface PD-L1 up-regulation during the UPR by either pharmacological or genetic means indicates that the IRE1α/XBP1 axis functions as a gatekeeper of PD-L1 expression in macrophages independently of IFNγ produced locally by T cells. By comparing gene expression in *Ern1-* and *Xbp1-*CKO macrophages, it became apparent that *Ern1* but not *Xbp1* regulates a UPR-mediated PD-L1 gene expression.

Mouse studies showed that the sensitivity to PD-L1 blockade depends on PD-L1 expression in myeloid cells (macrophages and dendritic cells) and not on tumor cells [69,70]. Remarkably, a recent report showed that ISRIB (integrated stress response inhibitor), a small molecule that reverses the effects of eIF2α phosphorylation downstream of PERK, reduces the abundance of the PD-L1 protein in murine *MYCTg;KRASG12D* liver cancer cells [51]. Whereas both reports agree on the role of the UPR in regulating PD-L1 expression, the discrepancy between our study and the previous study creates an interesting conundrum. Significantly, we found that in tumor-infiltrating macrophages isolated from human endometrial and breast cancers, the IRE1α gene signature is a better predictor of *CD274* (PD-L1) transcription than the PERK gene signature, confirming the conclusion reached in mouse macrophages. This suggests that IRE1α is an important, IFNγ-independent regulator of *CD274* in macrophages in cancer. Collectively, these results and considerations suggest that PD-L1 may be regulated by different arms of the UPR depending on the cell type. Future studies may better delineate these distinctions. Furthermore, since PD-L1 serves as the ligand for PD-1[+] T cells with an exhausted [71] or a regulatory phenotype [72], a plausible conclusion from the present study is that IRE1α inhibition in tumor-infiltrating myeloid cells could be used therapeutically to ameliorate the effects of immune dysregulation in the TME, including the down-regulation of PD-L1. This effect may work in concert with a mitigation of the IRE1α/XBP1 axis in T cells [67] to rescue a failing immune surveillance and restore immune competence locally.

A RIDD analysis in *Ern1-*deficient macrophages showed a dramatic loss of the integrity and connectivity of RIDD genes compared to control (*Ern1 fl/fl*) macrophages. This provides initial mechanistic evidence that RIDD may be involved in shaping the immune landscape in the TME, including PD-L1 expression. A possibility is that upon IRE1α activation, RIDD degrades not only mRNAs but microRNAs (miRNAs) as well, among which is miR-34a [73,74], a miRNA also shown to target *CD274* (PD-L1) mRNA by directly binding to its 3′-UTR [75,76]. The loss of RIDD integrity shown here suggests that RIDD/miR-34a could represent the link between IRE1α and *CD247* gene expression. Future studies will need to address the role of RIDD in PD-L1-driven immune dysregulation in the TME.

In conclusion, we provide evidence in support of UPR-driven mechanisms as a source of immune dysregulation in the TME. We have identified the IRE1α/XBP1 axis as a critical signaling pathway in macrophage polarization to a mixed IIS phenotype, PD-L1 expression, and tumor growth. Cell-nonautonomous IRE1α-dependent signaling has been proposed as a regulator of immune activation [77] and stress resistance and longevity in *Caenorhabditis elegans* [78], suggesting that the IRE1α/XBP1 axis may be central to intercellular communication during cellular stress. Here we further validate the view that UPR signals in the TME directly affect tumor-infiltrating macrophages promoting a complex immune dysregulation and defective tumor control in vivo. The fact that the IRE1α/XBP1 axis also regulates PD-L1 expression

points to the UPR as a general mechanism for immune dysregulation at the tumor and immune cells interface with myeloid cells, ultimately impairing the function of tumor-specific T cells [28,36] with loss of local immune surveillance.

# Materials and methods

## Cell lines and cell culture

Human cells lines colon carcinoma DLD1 and prostate PC3 and murine cell lines prostate TC1 and melanoma B16.F10 cancer cells were grown in RPMI or DMEM (Corning) supplemented with 10% FBS (HyClone) and 1% penicillin/streptomycin/L-glutamine, NEAA, sodium pyruvate, HEPES. All cells were maintained at 37˚C incubation with 5% $O_2$. All cell lines were mycoplasma free as determined PCR assay (Southern Biotech).

## Mice

APC mice were provided as a kind gift from Dr. Eyal Raz (UCSD). LysM-Cre mice were kindly provided by Dr. Richard Gallo (UCSD). ERN1[fl/fl] and XBP1[fl/fl] mice were kindly provided by Dr. Jonathan Lin (UCSD), who originally obtained them from Drs. Laurie Glimcher (Dana Farber, Harvard University) and Takao Iwawaki (RIKEN, Japan). All mice were housed in the UCSD vivarium according to approved protocols and animal welfare standards. Genotypes of CKO mice were confirmed by PCR on tissue obtained by ear punch and digested according to a standard protocol.

## TERS CM generation

DLD1 cells were induced to undergo ER stress through treatment of 300 nM Tg (Enzo Life Sciences) for 2 hours. Control cells were similarly treated with an equal volume of vehicle (0.02% ethanol). Cells were washed twice with Dulbecco's PBS (Corning) and then incubated in fresh, standard growth medium for 16 hours. CM was then harvested, centrifuged for 10 minutes at 2,000 RPM, filtered through a 0.22-μm filter (Millipore), and treated to cells or stored at −80˚C until use. For TERS priming, CM was generated from homologous cell type unless otherwise specified. To measure IFNγ in TERS CM, QBeads (Intellicyt, Ann Arbor, MI) were used following manufacturer's instructions. IFNγ was quantified on the iQue Screener PLUS (Intellicyt) using a standard cursive and manufacturer-provided template for analysis.

## BMDM and BMDC generation in culture

BMDCs were procured by isolating the femur and tibia of specified host and flushing out the bone marrow using cold, unsupplemented RPMI growth media (Corning) using a 27-gauge needle and syringe. Hemolysis was performed using ACK Lysis buffer (Bio Whittaker). For macrophage differentiation, bone marrow cells were incubated 1 week in standard growth medium supplemented with 30% L929 CM (LCM) or m-CSF (origin) at concentration.

## ERAI activity assay

Cancer cell line reporter cells were transduced with the ERAI construct, originally described [41]. Briefly, the pCAX-F-XBP1ΔDBD-venus (a kind gift from Dr. Iwawaki, Gunma University) underwent PCR using following primers: F: ctaccggactcagatctcgagccaccATGGACTA-CAAGGACGACG, R: gaattatctagagtcgcggccgcTTACTTGTACAGCTCGTCC. PCR fragments were cloned into pLVX-puro (Clontech) lentivirus vector with Gibson Assembly Mixture (NEB) according to manufacturer's instruction. Stbl3 competent cells were transformed to produce the plasmid insert, whose presence was confirmed by sequencing. For production of

lentivirus, 293FT (Invitrogen) cells were seeded in a 10-cm dish and transfected with a plasmid mixture of ERAI plasmid and psPAX2 and pMD2G viral packaging plasmids. The supernatant of virus-producing transfected cells was collected every 24 hours for 3 days posttransfection. Viral supernatant was concentrated by 10% PEG-8000 and pelleted with 2,000$g$ for 40 minutes at 4˚C and resuspended in PBS. Target cancer cells were transduced with lentivirus by supplementing with polybrene (8 μg/mL) to virus containing solution and loaded onto B16.F10 cancer cell line. Lines were transduced for 48 hours. Then, cells were washed twice with PBS and positively selected for using puromycin (2 μg/mL) for 2 weeks. In some instances, positively transduced cells were then stimulated for Venus expression and were sorted by FACS (BD) to isolate high-expressing clones. Lines were maintained under puromycin.

## Flow cytometry

Single-cell suspensions of myeloid cells were separated and stained for CD80 (B7-1) (BD Biosciences), PD-L1 (CD274) (BD Biosciences), and CD86 (BD Biosciences). Viable cells were determined by 7AAD exclusion, and data were acquired using a FACScalibur flow cytometer (BD). Flow results were analyzed using CellQuest Pro (BD) and Flow JO (Tree Star) software.

## RT-qPCR

mRNA was harvested from cells using Nucleopsin II Kit (Machery-Nagel) or enzymatically using the Zygem RNAgem Tissue PLUS kit (Microgembio, New Zealand). Concentration and purity of RNA were quantified the NanoDrop (ND-1000) spectrophotometer (Thermo Scientific) and analyzed with NanoDrop Software v3.8.0. RNA was normalized between conditions and cDNA generated using the High Capacity cDNA Synthesis kit (Life Technologies). RT-qPCR was performed on ABI 7300 Real-Time PCR system using TaqMan reagents for 50 cycles using universal cycling conditions. Cycling conditions followed manufacturer's specifications (KAPA Biosystems). Target gene expression was normalized to *β-actin* and relative expression determined by using the–ΔΔCt relative quantification method. Primers for RT-qPCR were purchased from Life Technologies: Arg1, (Mm00475988_m1), Cd274 (Mm03048248_m1), Chop (Mm00492097_m1), Grp78 (Mm00517691_m1), Il6 (Mm99999064_m1), Il23-p19 (Mm00518984_m1), and Tapbp (Mm00493417_m1).

## Western blot analysis

After treatment, cells were washed with ice-cold PBS and suspended in the RIPA Lysis Buffer system: 1X RIPA buffer and cocktail of protease inhibitors (Santa Cruz Biotechnology). Cell lysates were centrifuged at 16,000$g$ for 15 minutes and the supernatants were extracted. Protein concentration was determined using Pierce BCA Protein Assay Kit (Thermo Scientific). Samples were heat-denatured and equal concentrations of protein were electrophoresed on 4–20% Mini-PROTEAN TGX Precast Gels (Bio-Rad) and transferred onto 0.2-μm PVDF membrane in Tris-Glycine transfer buffer containing 20% methanol. The membranes were blocked with 5% nonfat milk in TBS containing 0.1% Tween-20 (TBS-T) for 1 hour at room temperature and subsequently incubated with diluted primary antibodies overnight at 4˚C. Membranes were washed for 5 minutes at room temperature 3 times by TBS-T, incubated with secondary antibody conjugated with horseradish peroxidase (HRP) in 5% nonfat milk for 1 hour at room temperature, and washed for 5 minutes at room temperature 3 times by TBS-T. Immunoreactivity was detected by chemiluminescence reaction using Pierce ECL Blotting Substrate (Thermo Scientific). Primary antibodies used were rabbit monoclonal antibody to IRE1α (clone 14C10) (Cell Signaling Technology), rabbit polyclonal antibody to XBP-1s (#83418) (Cell Signaling Technology), and goat polyclonal antibody to GAPDH (A-14) (Santa Cruz

Biotechnology). Bound primary antibodies were revealed by the following secondary antibodies: HRP-conjugated goat antibody to rabbit IgG (Cell Signaling Technology) and HRP-conjugated donkey antibody to goat IgG (sc2020) (Santa Cruz Biotechnology).

## Tumor studies

For orthotropic tumor implantation model, B16.F10 cancer cells ($n$ = 4) were detached from plastic, washed twice with cold PBS, and resuspended at a concentration of 300,000 cells/ml in PSB. Host C57BL/6 or transgenic ERAI mice (kindly provided by Dr. T. Iwawaki) were subcutaneously injected with 100 μl (30,000 cells) of cell suspension into the right hind flank. After approximately 22 days, mice bearing tumors greater than 1 cm were euthanized. For tumor growth studies, B16.F10 were subcutaneously injected in C57BL/6 (WT) or TLR4-KO mice (a kind gift from Dr. M. Corr [UCSD]). Tumor establishment was first determined by palpation and size was then measured in two dimensions using calipers. When tumors reached >20 mm in any one dimension or after 30 days postimplantation, whichever came first, mice were euthanized. Tumor volume was calculated using the ellipsoid volume formula, $V = 1/2$ $(H \times W^2)$. All mice were euthanized when any tumor reached 20 mm in any one dimension, per UCSD animal welfare standards, or after 30 days postimplantation. Tumor volume was calculated using the ellipsoid formula: $V = 1/2 (H \times W^2)$.

## Isolation of CD11b$^+$ and F4/80 cells

For the B16.F10 model, B16.F10 cancer cells ($n$ = 5) were subcutaneously injected (30,000 cells) into the right hind flank of C57BL/6 mice. After approximately 22 days, mice bearing tumors greater than 1 cm were euthanized. For the APC model, APC mice were genotyped for *APC* mutation to confirmed homozygosity of transgene. At approximately 12–15 weeks of age, APC mice were euthanized by cervical dislocation. The small intestine was removed from host and cut longitudinally, running parallel to the intestinal lining. Adenomas lining the intestine were excised using an open blade and pooled, respective to the host, in ice-cold PBS supplemented with 0.5% (w/v) bovine serum albumin (BSA). For both model systems, once the tumor, spleen, and bone marrow were isolated from tumor-bearing hosts, tissues were dissociated through enzymatic digestion (TrypLE) at 37˚C for 30 minutes on a rocker 85 plate, followed by cell straining through a 22-μm filter in ice-cold PBS + 0.5% (w/v) BSA. Cell suspensions were then stained for CD11b$^+$ positivity by first using a CD11b-biotin conjugated antibody (BD Biosciences) and incubated for 15 minutes at 4˚C. Cells were then washed twice with PBS + 0.5% BSA and positively selected by magnetic separation using a biotin isolation kit (Stem Cell) according to manufacturer's specifications. F4/80$^+$ macrophages were isolated from subcutaneous B16.ERAI tumors from the right hind flank *Ern1 x LysMCre* or *fl/fl* mice. After approximately 22 days, mice bearing tumors >1 cm in length were euthanized. Tumors and spleens were isolated, and tissues were dissociated through enzymatic digestion (TrypLE) at 37˚C for 30 minutes on a rocker 85 plate, followed by cell straining through a 22-μm filter in ice-cold PBS + 0.5% (w/v) BSA. Cell suspensions were then stained for F4/80$^+$ positivity by first using a F4/80-PE-conjugated antibody (StemCell Technologies Cat# 60027PE.1) and incubated for 15 minutes at 4˚C. Cells were then washed twice with PBS 0.5% BSA and positively selected by magnetic separation using PE Positive Selection Kit II (StemCell Technologies) according to manufacturer's specifications.

## RNASeq analysis

RNA was extracted from wild-type or Ern1-CKO BMDMs that were untreated or treated with TERS CM for 18 hours using the Nucleospin RNA kit (Macherey Nagel). Each group consisted

of three independently derived BMDM cultures. RNA sample purity was ascertained by the NanoDrop quantification method. Single-end stranded RNA libraries were sequenced on an Illumina HiSeq 4000. All samples and replicates were sequenced together on the same run. All 12 mouse RNASeq transcript quantification was performed with sailfish version 0.9.2 [79], using the GRCm38 mouse transcriptome downloaded from Ensembl (ftp://ftp.ensembl.org/pub/release-97/fasta/mus_musculus/cdna/Mus_musculus.GRCm38.cdna.all.fa.gz) with default values. The 33 RIDD target genes were collected from [49]. We *z* scored these RIDD target genes within each group separately (*Ern1* fl/fl and *Ern1* CKO), and then mean value was calculated and compared between different phenotype (untreated vs TERS CM treated) within each group.

## OLS linear model predicting PD-L1 using IRE1α pathway and PERK pathway downstream genes

OLS models were fitted and compared using the Python (version 2.7.15) statsmodels package (version 0.9.0). We collected IRE1α pathway (R-HSA-381070.1) and PERK pathway (R-HSA-381042.1) downstream genes from REACTOME [54]. Each gene was *z* scored to ensure a mean of 0 and standard deviation of 1. Because quantification of transcript levels is noisier when genes are expressed at low levels, we implemented a filter to remove genes expressed under a certain threshold and evaluated pathway scores at thresholds ranging from 0 to 1,000 reads. We then fitted models at different thresholds to evaluate robustness of the model to choice of threshold. Models were fitted using the formula:

$$PD - L1 = \beta_0 + \Sigma \beta_i \cdot gene_i$$

Nested OLS models with ERN1 only and ERN1 + PERK were compared using the AIC. For each model, the AIC was calculated as AIC = $2k - 2\ln(L)$, where $k$ represents the number of estimated values, and $L$ represents the likelihood function for the model. Models were compared using the formula $\exp((AIC_{min} - AIC_i)/2)$, which represents the relative likelihood of model i with respect to the best available model.

## Statistical analysis

To determine if differences between groups were statistically significant for PCR experiments, groups were compared using unpaired Student's *t* tests with Welch's correction. Statistically significant differences are indicated as follows: $^*p < 0.05$, $^{**}p < 0.01$, $^{***}p < 0.001$, $^{****}p < 0.0001$. Statistical significance in tumor growth experiments was determined using the Mann-Whitney *t* test and survival curves were generated by the Kaplan-Meier method.

## Supporting information

**S1 Fig.** (A). Flow cytometry analysis of CD11b[+] cells in the spleen, the BM, and within the TME of B16.F10 tumors in C57BL/6 mice carrying the *Xbp1-Venus* fusion transgene. (B) Analysis of F4/80 expression on CD45+ cells in B16.F10 tumors. BM, bone marrow proximal to the tumor; TME, tumor microenvironment.
(PDF)

**S2 Fig.** Dose-dependent 4μ8C-mediated inhibition of ERAI induction by TERS CM in B16. F10 melanoma cells (A) and quantification of 4μ8C inhibition of Xbp1 splicing in C57BL/6 mice macrophages stimulated with TERS CM (B). Data are included in S2 Data. ERAI, ER stress-activated indicator; TERS CM, transmissible ER stress conditioned medium; XBP1, X-

box binding protein 1.
(PDF)

**S3 Fig. Chemical inhibition of PERK signaling does not affect IIS polarization of BMDM in vitro.** (A) Western blot analysis for pPERK in whole-cell lysates from BMDM treated with Tg with or without 4μ8C (30 μM) or GSK2656156 (10 nM). (B) Expression of selected genes by RT-qPCR by mRNA from BMDM cultured in TERS CM or in vehicle Veh CM with or without GSK2656157 (50 nM) ($n$ = 4). Error bars represent SEM. (C) Surface expression (flow cytometry) of CD86 and PD-L1 in BMDM cultured in TERS CM or in vehicle Veh CM with or without GSK2656157 (50 nM). Data are included in S2 Data. BMDM, bone marrow–derived macrophage; CM, conditioned medium; IIS, proinflammatory/immune-suppressive; PD-L1, programmed death ligand 1; PERK, PKR-like ER kinase; pPERK, phosphorylated PERK; RT-qPCR, reverse transcriptase quantitative PCR; TERS CM, transmissible ER stress CM; Tg, thapsigargin.
(PDF)

**S4 Fig. BMDMs were generated from wild-type C57BL/6 mice were untreated or treated with 4HNE (30μM), LPS (100 ng/ml), and lactic acid (30 mM) for 1, 6, or 24 hours in the absence or presence of 4μ8C (30 μM).** At the indicated time points, RNA was isolated using Nucleospin 2 kit and processed for RT-qPCR. Values represent the mean ± SEM ($n$ = 5 per group). Data are included in S2 Data. 4HNE, 4-hydroxynonenal; BMDM, bone marrow–derived macrophage; LPS, lipopolysaccharides; RT-qPCR, reverse transcriptase quantitative PCR.
(PDF)

**S5 Fig. Genotype analysis of wild-type (*fl/fl*) and *Ern1*- or *Xbp1*-CKO mice.** For each mouse, genomic DNA was extracted from an ear punch and subjected to 3 PCR experiments. The first PCR (upper panel) used primers designed to evaluate the floxed status of *Ern1* or *Xbp1*, with the floxed band appearing at 229 bp (*Ern1*) or 141 bp (*Xbp1*) and the wild-type band appearing at 254 bp (*Ern1*) or 183 bp (*Xbp1*). The band at approximately 200 bp in *Ern1* is nonspecific. The second PCR (middle panel) used primers to detect the presence of the Cre insertion following the LysM promoter, with the Cre insertion appearing at approximately 700 bp. The band at 350 bp signifies the LysM promoter without Cre insertion (wild type). The third PCR (lower panel) used primers specific for the wild-type LysM promoter (without Cre), which appears 350 bp. CKO, conditional knock-out; *Xbp1*, X-box binding protein 1.
(PDF)

**S6 Fig. QBeads were used to measure IFNγ in control CM or in two independently generated batches of TERS CM.** The standard curve provided by the manufacturer was used to quantify each sample (A). BMDMs generated from wild-type C57BL/6 mice were untreated or treated with TERS, with and without a blocking antibody for IFNγ for 18 hours. RNA was isolated using Nucleospin 2 kit and processed for RT-qPCR (B). Boxplot showing the *IFNG* gene expression in Ern1(fl/fl) and Ern1 LysMCre groups from the RNASeq data set (C). Data are included in S2 Data. BMDM, bone marrow–derived macrophage; CM, conditioned medium; IFNγ, interferon gamma; RNASeq, RNA sequencing; RT-qPCR, reverse transcriptase quantitative PCR; TERS CM, transmissible ER stress CM.
(PDF)

**S7 Fig.** RNASeq analysis of *Tapbp* expression in untreated or TERS CM–treated wild type and *Ern1*-CKO BMDM (A). RT-qPCR analysis of *Tapbp* expression analysis in untreated or TERS CM–treated wild type and *Xbp1*-CKO BMDM (B). Data are included in S2 Data. BMDM,

bone marrow–derived macrophage; CKO, conditional knock-out; RNASeq, RNA sequencing; *Tapbp*, tapasin; RT-qPCR, reverse transcriptase quantitative PCR; TERS CM, transmissible ER stress conditioned medium; *Xbp1*, X-box binding protein 1.
(PDF)

**S8 Fig. Spearman correlation analysis showing a higher power of *ERN1* in bulk tumor sequencing in predicting *CD274* expression when macrophage infiltration is high.** (A) Spearman correlation between *ERN1* expression and *EIF2AK3* expression from TCGA pancancer study ($n$ = 9,607). Both genes are normalized to TPM and in log2 scale. (B) Spearman correlation between *ERN1* expression and CD274 expression from TCGA pancancer study ($n$ = 9,607). Red dots are samples with high macrophage infiltration scores ($>$70%), and blue dots are samples with low macrophage infiltration scores ($<$30%). (C) Spearman correlation between EIF2AK3 expression and CD274 expression from TCGA pancancer study ($n$ = 9,607). Red dots are samples with high macrophage infiltration scores ($>$70%), and blue dots are samples with low macrophage infiltration scores ($<$30%). Data are included in S2 Data. EIF2AK3, translation initiation factor 2; TCGA, The Cancer Genome Atlas; TPM, transcripts per million.
(PDF)

**S9 Fig. List of genes used in the aggregate pathway score for the IRE1α and PERK pathway after filtering.** Black stands for the original gene sets. Blue and yellow colored genes are used in the aggregate pathway score after filtering out genes with less than 500 and 1,000 read counts, respectively. IRE1α, inositol-requiring enzyme 1; PERK, PKR-like ER kinase.
(PDF)

**S10 Fig. Chemical inhibition of IRE1α but not PERK signaling affects *Vegf* gene transcription in BMDM in vitro.** Expression of *Vegf* by RT-qPCR by mRNA from BMDM cultured for 18 hours in TERS CM or in vehicle Veh CM with or without 4µ8C (30 µM) ($n$ = 3) or GSK2656157 (10 nM) ($n$ = 2). Error bars represent SEM. Data are included in S2 Data. BMDM, bone marrow–derived macrophage; CM, conditioned medium; IRE1α, inositol-requiring enzyme 1; PERK, PKR-like ER kinase; RT-qPCR, reverse transcriptase quantitative PCR; TERS CM, transmissible ER stress CM; *Vegf*, vascular endothelial growth factor.
(PDF)

**S1 Data. Supporting Data for primary Figs 1–6.**
(XLSX)

**S2 Data. Data for Figures in Supporting Information.**
(XLSX)

## Acknowledgments

The authors thank Valentina Ferrari for performing the QBeads assay.

## Author Contributions

**Conceptualization:** Jeffrey J. Rodvold, Jonathan Lin, Kristen Jepsen, Hannah Carter, Maurizio Zanetti.

**Formal analysis:** Alyssa Batista, Jeffrey J. Rodvold, Su Xian, Stephen C. Searles, Gonzalo Almanza, T. Cameron Waller, Hannah Carter, Maurizio Zanetti.

**Investigation:** Alyssa Batista, Jeffrey J. Rodvold, Su Xian, Stephen C. Searles, Alyssa Lew, Gonzalo Almanza, Kristen Jepsen.

**Methodology:** Gonzalo Almanza.

**Resources:** Takao Iwawaki, Jonathan Lin.

**Writing – original draft:** Maurizio Zanetti.

**Writing – review & editing:** Jeffrey J. Rodvold, Su Xian, Stephen C. Searles, T. Cameron Waller, Jonathan Lin, Kristen Jepsen, Hannah Carter, Maurizio Zanetti.

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
