## [Editor Report · Decision Letter 0]

12 Feb 2020

Dear Dr Zanetti, 

Thank you for submitting your manuscript entitled "IRE1� regulates macrophage polarization, PD-L1 expression and tumor survival" for consideration as a Research Article by PLOS Biology.

Your manuscript has now been evaluated by the PLOS Biology editorial staff as well as by an academic editor with relevant expertise and I am writing to let you know that we would like to send your submission out for external peer review.

Please re-submit your manuscript within two working days, i.e. by Feb 14 2020 11:59PM.

Kind regards,

Di Jiang,

Associate Editor

PLOS Biology

---

## [Decision Letter · Decision Letter 1]

18 Mar 2020

Dear Dr Zanetti,

Thank you very much for submitting your manuscript "IRE1a regulates macrophage polarization, PD-L1 expression and tumor survival" for consideration as a Research Article by PLOS Biology. Your manuscript was evaluated by the PLOS Biology editors as well as by an Academic Editor with relevant expertise and by two independent reviewers. The reviewers appreciated the attention to an important topic. 

Based on the reviews, we will probably accept this manuscript for publication, assuming that you will modify the manuscript to address all the points raised by the reviewers. Please also make sure to address the data and other policy-related requests noted at the end of this email.

We expect to receive your revised manuscript within two weeks. Your revisions should address the specific points made by each reviewer. In addition to the remaining revisions and before we will be able to formally accept your manuscript and consider it "in press", we also need to ensure that your article conforms to our guidelines. A member of our team will be in touch shortly with a set of requests. As we can't proceed until these requirements are met, your swift response will help prevent delays to publication.

*Copyediting*

*Published Peer Review History*

*Early Version*

*Submitting Your Revision*

Sincerely,

Di Jiang, PhD

Associate Editor

PLOS Biology

FINANCIAL STATEMENT:

-- Please provide a number for the grant from Eva B. Buck Foundation.

DATA POLICY:

-- Regardless of the method selected, please ensure that you provide the individual numerical values that underlie the summary data displayed in the following figure panels as they are essential for readers to assess your analysis and to reproduce it: Figures 1A-E, 2A-D, 3DE, 4AB, 5A-E, 6AB, S2A, S3BC, S4, S6A-C, S7AB, S8A-C, S10. NOTE: the numerical data provided should include all replicates AND the way in which the plotted mean and errors were derived (it should not present only the mean/average values).

-- Please deposit the RNASeq data in a public repository, arrange it to be accessible to the public at the time of publication of this manuscript, and please provide us with an editor/reviewer key or token so that we can check the data before we accept the paper. 

Reviewer remarks:

Reviewer #1 (Eric Chevet, signed review): 

Manuscript PBIOLOGY-D-20-00290R1 entitled 'IRE1 regulates macrophage polarization, PD-L1 expression and tumor survival' by Batista and colleagues for publication in PLOS Biology.

In this manuscript Batista et al. demonstrate IRE1/XBP1s-dependent regulation of tumor-associated macrophages polarization though at least in part the regulation of PDL1 molecule. The authors provide very robust and convincing data using mouse models and human datasets. Hereby the authors document the links between UPR/IRE1 and macrophage polarization that contribute to cancer immunosuppression. The authors could improve their manuscript by addressing the following minor points:

- the authors should comment on the variability of the samples tested in figure 1 B to E; 3E; 5A and 5D. 

- total PERK expression is missing in Figure S3A.

- the authors should define the abbreviations used in Table 1.

- the authors should correct the typos:

Page 8: 'Grp78 instead of Gr78'

Figure 3: Chaperones Cheparones 

- the authors further should comment the discrepancy between the results and in regards to PERK involvement in the PDL1 regulation. Could the data presented in Table 1 help for this ?

Reviewer #2 (Michael McDermott, signed review): 

This is a well written paper, with clear background, carefully controlled experiments with interesting data, and adequate references. The authors have examined the mechanisms whereby the UPR may regulate immune cells by perturbing the TME and increasing tumor cell invasiveness. They report that significantly reduced polarization of mouse macrophagse, and also reduced CD86 and PD-L1 expression using genetic deletion of the IRE1Xbp1 axis and also by pharmacological blockade of this axis. 

Mice with IRE1-deficient macrophages had significantly greater survival than controls when implanted with B16.F10 melanoma cells. By contrast this was not found with Xbp1-depletion and suggests a novel mechanism in PD-L1 regulation in macrophages. 

They also showed, using RNASeq analysis, that BDMS in which the IRE1gene was deleted by conditional KO, lost the integrity of the RIDD gene network and were also unable to activate CD274 gene expression. These effects were cell specific as they were not seen in tumour cells. It is suggested that IRE1 inhibition in TIMs could help overcome PD-L1 induced immune dysregulation in tumors and restore immune surveillance. 

In summary the IRE1 axis is crucial for macrophage function and polarisation, in the tumor microenvironment, and this may occur independently of IFN signalling. However, further work is necessary to assess the role of RIDD in PD-L1-driven immunoregulation. 

Suggestions.

 In the discussion the authors might consider add a bit more detail on the potential of IRE1 inhibition in TIMs e.g which tumors, and under which circumstances? Also, they might discussion the consequences or IRE1a inhibition; it has been shown that the IRE1α-XBP1 pathway controls T cell function in ovarian cancer by regulating mitochondrial activity (Song M, et al. Nature. 2018; 562:423-8. doi: 10.1038/s41586-018-0597-x). An overactive IRE1α-XBP1 pathway may reprograms M1 macrophages toward an increased metabolic state, in conditions like Cystic Fibrosis, with increased glycolytic rates and mitochondrial function, and exaggerated production of TNF and IL-6. Significantly higher levels of XBP1s protein are present, in CF M1 macrophages, while XBP1s was not detected in M2 macrophages. A reference to the role of altered metabolic state in macrophage polarization would expand the discussion (Lara-Reyna S, et al. Front Immunol. 2019;10:1789).

---

## [Editor Report · Decision Letter 2]

20 May 2020

Dear Dr Zanetti,

On behalf of my colleagues and the Academic Editor, Thomas C. Freeman, I am pleased to inform you that we will be delighted to publish your Research Article in PLOS Biology. 

Early Version

PRESS 

Kind regards,

Vita Usova

Publication Assistant, 

PLOS Biology

on behalf of

Di Jiang,

Associate Editor

PLOS Biology